# Elevated MACC1 Expression in Colorectal Cancer Is Driven by Chromosomal Instability and Is Associated with Molecular Subtype and Worse Patient Survival

**DOI:** 10.3390/cancers14071749

**Published:** 2022-03-29

**Authors:** Vincent Vuaroqueaux, Alexandra Musch, Dennis Kobelt, Thomas Risch, Pia Herrmann, Susen Burock, Anne-Lise Peille, Marie-Laure Yaspo, Heinz-Herbert Fiebig, Ulrike Stein

**Affiliations:** 14HF Biotec GmbH, Am Flughafen 14, 79108 Freiburg, Germany; vincent.vuaroqueaux@4hf.eu (V.V.); alexandra.musch@4hf.eu (A.M.); anne-lise.peille@4hf.eu (A.-L.P.); fiebig@4hf.eu (H.-H.F.); 2Experimental and Clinical Research Center, Charité–Universitätsmedizin Berlin and Max-Delbrück-Center for Molecular Medicine, 13125 Berlin, Germanypia.herrmann@charite.de (P.H.); 3German Cancer Consortium (DKTK), 69120 Heidelberg, Germany; 4Otto Warburg Laboratory “Gene Regulation and Systems Biology of Cancer”, Max Planck Institute for Molecular Genetics, 14195 Berlin, Germany; risch@molgen.mpg.de (T.R.); yaspo@molgen.mpg.de (M.-L.Y.); 5Charité Comprehensive Cancer Center, 10117 Berlin, Germany; susen.burock@charite.de

**Keywords:** CRC, MACC1, gene copy number alteration, gene expression, survival

## Abstract

**Simple Summary:**

Elevated expression of Metastasis-Associated in Colon Cancer 1 (*MACC1*) has been identified as a strong prognostic marker of adverse outcomes for human colorectal (CRC) and other solid cancers. The biological basis of high *MACC1* expression and the context of its occurrence are still poorly understood. This study investigated whether chromosomal instability and somatic copy number alterations (SCNA) frequently occurring in CRC contribute to *MACC1* dysregulation, with prognostic and predictive impact.

**Abstract:**

Metastasis-Associated in Colon Cancer 1 (*MACC1*) is a strong prognostic biomarker inducing proliferation, migration, invasiveness, and metastasis of cancer cells. The context of *MACC1* dysregulation in cancers is, however, still poorly understood. Here, we investigated whether chromosomal instability and somatic copy number alterations (SCNA) frequently occurring in CRC contribute to *MACC1* dysregulation, with prognostic and predictive impacts. Using the Oncotrack and Charité CRC cohorts of CRC patients, we showed that elevated *MACC1* mRNA expression was tightly dependent on increased *MACC1* gene SCNA and was associated with metastasis and shorter metastasis free survival. Deep analysis of the COAD-READ TCGA cohort revealed elevated *MACC1* expression due to SCNA for advanced tumors exhibiting high chromosomal instability (CIN), and predominantly classified as CMS2 and CMS4 transcriptomic subtypes. For that cohort, we validated that elevated *MACC1* mRNA expression correlated with reduced disease-free and overall survival. In conclusion, this study gives insights into the context of *MACC1* expression in CRC. Increased *MACC1* expression is largely driven by CIN, SCNA gains, and molecular subtypes, potentially determining the molecular risk for metastasis that might serve as a basis for patient-tailored treatment decisions.

## 1. Introduction

Metastatic dissemination of primary tumors is directly linked to patient survival representing the most lethal attribute of cancer. It critically limits successful therapy in many tumor entities. Biomarkers identifying cancer patients at high risk for metastasis and simultaneously acting as key drivers for metastasis are extremely desired. Until today, about 25–30% of all colorectal cancer (CRC) patients are already distantly metastasized when presenting the first time (stage IV); 40–50% of all patients newly diagnosed with CRC without distant metastases (stages I–III) will develop distant metastasis later (metachronously) after primary surgery, which is significantly linked to shorter survival. Patient survival is about 80% in the early stages, but below 10% when distant metastases occurred [1,2,3]. Thus, novel prognostic tools for patient stratification identifying those patients at high risk for distant metastasis in early stages followed by tailored novel therapeutic options are ultimately desired.

We discovered the novel, previously undescribed gene Metastasis-Associated in Colon Cancer 1 (*MACC1*) in human CRC in our group [4]. MACC1 induces fundamental processes, such as proliferation, migration, invasiveness, and metastasis formation in xenografted and transgenic mice [5,6]. Meanwhile, MACC1 has been established by us and many groups as a key player, and prognostic and predictive biomarker for tumor progression and metastasis in more than 20 solid cancer entities, including CRC, with meta-analyses to solid cancers, hepatocellular cancer, and gastrointestinal tract cancers, such as CRC and gastric cancer [7,8,9,10,11,12].

High *MACC1* expression levels, determined in the primary tumor or in cancer patient blood, predict tumor aggressiveness and metastasis formation linked to shorter patient survival [6,13]. Identification of the *MACC1* gene promoter has unveiled transcription factors and respective transcription factor binding sites regulating the transcription of the metastasis inducer *MACC1* [14,15]. Post-transcriptional regulations of *MACC1* expression by miRNA, lnc-RNA and circRNA are also reported [5,16].

Genetic factors might provide additional layers for *MACC1* expression regulation resulting in phenotypical features of cell proliferation, dissemination, and motility, and more importantly link to tumor progression, metastasis formation, treatment response, and ultimately, to patient survival.

For CRC, single nucleotide polymorphisms (SNPs) were reported in intronic regions 1, 2 and 6 as well as in the coding region of *MACC1* [17,18,19]. Some of these SNPs were found to be clinically relevant and were linked to shorter patient survival as well as increased risk for metachronous metastasis for patients younger than 60 years with stage I or II CRC tumors (rs1990172, rs975263) or increased risk of recurrence after liver transplantation (rs1990172, rs975263). Further clinically relevant *MACC1* SNPs were identified for breast cancer [20,21,22] and hepatocellular carcinoma [23]. Besides SNPs, somatic copy number alteration (SCNA) might have a prognostic and predictive impact for *MACC1* expression regulation and function. Here, we hypothesized a potential correlation of SCNA of *MACC1* as a novel regulatory category of *MACC1* expression levels and association with molecular CRC subtypes. Therefore, we proved independent CRC patient cohorts to determine whether high *MACC1* expression was associated with genomic alteration events occurring in CRC tumors, whether they could predict metastasis formation in early-stage patients. Next, by using the TCGA colon and rectum adenocarcinoma (COAD-READ) cohorts, we have refined the context of *MACC1* expression with CRC molecular subtypes and the association with cancer patients’ clinical outcomes. Altogether, these analyses contribute to a better understanding of *MACC1* expression modalities in CRC and their impact on patient prognosis.

## 2. Materials and Methods

Information about the CRC cohorts used is presented in Appendix A.

### 2.1. Oncotrack Cohort of CRC Patients

In this cohort, tumor tissue samples of 106 CRC patients collected by the Oncotrack consortium (RRID:SCR_003767) were analyzed. All experiments were performed in accordance with the guidelines approved by the institutional review boards, number EA 1/069/11, of the Charité–Universitätsmedizin Berlin, Germany, and the ethics committee of the Medical University Graz, confirmed by the ethics committee of the John of God Hospital Graz (23-015 ex 10/11). All patients gave their written consent. The authors complied with all relevant ethical regulations for research involving human participants. For detailed patient characteristics and for all methodological aspects, please see Schütte et al. [24].

### 2.2. SCNA and Gene Expression Analysis of the Oncotrack Cohort

The OncoTrack CRC cohort was analyzed as previously published [25]. In brief, *MACC1* SCNA was estimated from whole genome or exome sequencing data. DNA reads were aligned to the human reference genome hg19 using BWA (bwa0.7.7-r441-mem for 75/101 bp, bwa0.5.9-r16-aln for 51 bp reads, RRID:SCR_010910). SCNA was estimated using the BICSeq algorithm and the read coverage data of tumor versus normal pairs [26]. We inferred ploidy using the B allele frequencies of heterozygous germline variants. For low coverage whole genome sequencing without matching blood data, we used as a proxy an electronic pool of six sex-matched normal samples.

For *MACC1* expression, RNA reads were aligned to hg19 using BWA and SAM tools (RRID:SCR_002105). Mapped reads were annotated using Ensembl v70 (RRID:SCR_002344). Gene expression levels were quantified in reads per kilobase of exon per million mapped reads (RPKM).

### 2.3. Charité Metastasis Cohort of CRC Patients

For this study, tumor tissue samples of 35 CRC patients were analyzed. All experiments were carried out in accordance with the guidelines approved by the institutional review board, number AA3/03/45, of the Charité–Universitätsmedizin Berlin, Germany. All patients gave their written consent (Charité–Universitätsmedizin, Berlin, Campus Mitte). The authors complied with all relevant ethical regulations for research involving human participants.

All patients were without familial history of CRC, did not receive any prior cancer treatment, and underwent R0 resection of the primary tumor (complete resection with no microscopic residual tumor). Snap frozen samples were subjected to molecular analysis. For detailed patient characteristics: see Appendix A.

### 2.4. DNA Isolation and ddPCR

For quantification of *MACC1* SCNA in tumor tissue of CRC patients from the Charité Metastasis cohort, genomic DNA was isolated using the Charge Switch gDNA Micro Tissue Kit (Thermo Fisher, Waltham, MA, USA) according to manufacturer’s instruction. DNA was dissolved in water and quantified using NanoDrop 2000 spectrophotometer (RRID:SCR_018042). Equal amounts were used for copy number analysis in droplet digital PCR (ddPCR). In brief, the reaction mix was prepared using 1x EvaGreen Supermix (Bio-Rad, Hercules, CA, USA), 100 nM of each primer (Biotez, Berlin, Germany)10 U BamHI (Fermentas now Thermo Fisher, Waltham, MA, USA) and 10 ng genomic DNA. The primers used are presented in Appendix A.

For droplet generation, the DG8 cartridges were filled with complete master mix and oil (Bio-Rad) at the respective positions, using the QX200 droplet generator (Bio-Rad, RRID:SCR_019707). The partitioned reaction mix containing up to 20,000 droplets was transferred carefully using a multichannel pipette equipped with filter tips (Rainin) to 96-well plates (Eppendorf, Germany). After initial denaturation at 95 °C for 10 min, 40 PCR cycles were performed in a T100 Thermal Cycler PCR machine (Bio-Rad) each built of 94 °C/30 s, 58 °C/30 s, and 98 °C/10 min. For quantification of PCR product positive droplets, the sealed plates were moved to the QX200 droplet reader (Bio-Rad, RRID:SCR_019707). Droplet counts were analyzed, and Poisson corrected using the QuantaLife software package (Bio-Rad).

### 2.5. TCGA COAD-READ Cohorts of CRC Patients

*MACC1* expression modalities in normal colon and in CRC were analyzed by using the COAD-READ dataset which was generated by The Cancer Genome Atlas Network (TCGA, (RRID:SCR_003193)). The TCGA data were downloaded from the Firehose platform (https://gdac.broadinstitute.org/ (accessed on 08 May 2018)). The collected cohort consists of a total of 632 cancer patients with clinical information (Appendix A). Most of the resected tumors, as well as normal tissues taken adjacent to the tumors, were analyzed for molecular features. A total of 431 samples had available RNA-Seq data, of which 380 were tumors and 51 were normal tissues; 598 tumor samples had SNP6.0 array profiling for SCNA and somatic mutations were available for 473 tumor samples obtained from paired tumor and normal tissue analyses by whole exome and whole genome sequencing. The study was extended by using the dataset reported by Wang et al. [27] and that merged the transcriptome of the COAD-READ cohort with those of the normal colon tissues from the Genotype-Tissue Expression (GTEx, (RRID:SCR_013042)) project. In depth molecular characteristics and subtypes of the COAD-READ samples and details about cancer patients were taken from the publication of Liu et al. [28].

### 2.6. Statistical Analysis

Statistical analyses were carried out using GraphPad software (RRID:SCR_002798) and the “R” statistical computing environment (RRID:SCR_001905) and associated modules from Bioconductor (RRID:SCR_006442), in the Linux operating system (Ubuntu, 12.04.5 [29]). The association between *MACC1* gene ploidy and expression, with clinical data (tumor location, grade, lymph node invasion or metastatic status) and molecular subtypes were conducted by using the Mann–Whitney test or Kruskal–Wallis test. The prognostic value of *MACC1* was assessed by univariate and multivariate Cox analysis against metastasis-free, disease-free and overall survival (MFS, DFS, OS), and in different patient subgroups according to the CRC subtype status. For all subsequent analyses, the median was used as the cut-off point for *MACC1* expression. Survival probabilities were calculated according to the Kaplan–Meier method, and group differences were assessed by the log rank test. Multivariate *p*-values were based on Wald statistics. Statistical analysis was performed with ”R” statistical software version 3.6.3 using the “survival” package (RRID:SCR_021137) [29].

### 2.7. Data Availability

The sources of the molecular data used in this study are indicated in Appendix A.

## 3. Results

### 3.1. Correlation between Increased MACC1 SCNAs and Elevated Expression Levels in CRC Patient Samples (Oncotrack Cohort)

We first tested for *MACC1* expression level dependency on gene SCNA in human tumors. Based on the Oncotrack cohort data (see material and methods) in which both *MACC1* SCNA and mRNA expression information was available for 90 tumors, we observed that *MACC1* expression levels were tightly copy number dependent (*p* = 0.002, Kruskal–Wallis test, excluding 1N as category due to small sample size) (Figure 1A). Compared to samples with no somatic copy number alteration (SCNA = 2N) there was an increase of *MACC1* expression level with increased SCNA that was further elevated with each gain of *MACC1* copy number. Surprisingly, one sample showed no *MACC1* expression after the loss of one genomic *MACC1* copy. The highest *MACC1* expression levels were found in human CRC with 4N and >4N. Thus, we identified an additional layer that regulates *MACC1* expression, potentially indicating the patient´s risk for metastasis formation. We further showed that the *MACC1* SCNA and elevated mRNA expression correlated with higher protein levels by using data issued from the proteogenomic study published by Vasaikar et al. [30] (Appendix A) (data and figure accessible via the cbio portal: https://www.cbioportal.org/, accessed on 08 March 2022).

### 3.2. Correlation between Increased MACC1 SCNAs, Elevated Expression Levels and Metastasis Formation in CRC Patients (Charité Cohort)

By using the Charité cohort (N = 35), we tested next for a correlation of CRC patients, who are low in MACC1 and have no distant metastasis, and those patients who are high in MACC1 and have metastasis, with respect to SCNA. Impressively, we found significantly higher SCNAs in the MACC1 high/with metastasis group vs. MACC1 low/no metastasis (Mann–Whitney *p* = 0.008, *t*-test *p* = 0.005; Figure 1B). The grouping in MACC1 low/high expressing patients was performed earlier by ROC analysis [4] (Figure 1C). The median metastasis-free survival SCNAs < cut-off vs. > cut-off is significantly different (*p* = 0.0035 log-rank (Mantel–Cox) and *p* = 0.002 Gehan–Breslow–Wilcoxon; cut-off = 3.34 copies) (Figure 1D). This cut-off was associated with a sensitivity of 77.78% and a specificity of 76.92%. For SCNA > cut-off patients, the metastasis-free survival was 42.47 months; for the SCNAs < cut-off patients, the metastasis-free survival was undefined (these patients do not have any metastasis). The longest follow-up time was 168 months (then the patient died), the shortest was 8 months. Taken together, by the inclusion of the MACC1 SCNA, we cannot only predict the MACC1 expression level but potentially also the distant metastatic capacity of CRC patients.

### 3.3. Correlation between Increased MACC1 SCNA, Elevated Expression Levels and Clinical, Molecular Subtypes, and Patient Survival Variables (TCGA COAD-READ Cohort)

#### 3.3.1. Association with Anatomopathological Parameters

We aimed to confirm that *MACC1* expression was dependent on tumor genetic background on a larger scale. We analyzed the modalities of *MACC1* mRNA expression within the COAD and READ cohorts from the TCGA (details of the cohorts: see material and methods and Appendix A). Overall, *MACC1* expression (RNA-Seq, RSEM normalized, log_2_ transformed; [31]) varied from 5.45 to 13.07. In both COAD and READ cohorts, tumors expressed higher *MACC1* levels than normal tissues (Wilcoxon test: COAD: *p* < 2.2 × 10^−16^, READ: *p* = 2.4 × 10^−7^) (Figure 2A). High *MACC1* expression in tumor samples was further confirmed with a derived dataset reported by Wang et al., 2018 [27] in which COAD-READ tumors were unified with 729 normal tissues from the GTEx database (https://gtextportal.org/home/, accessed on 08 May 2018) (median: COAD tumors = 6.7, and GTEx normal colons = 3.65; READ tumors = 7.14, GTEx normal rectums = 3.52, Wilcoxon test: both *p* < 2.2 × 10^−16^) (Appendix A). In both analyses, *MACC1* expression levels did not differ in normal samples from proximal and distal parts of the colon and rectum. *MACC1* level was found to be slightly higher in the READ cohort compared to COAD (median: READ = 10.7; COAD = 10.4, Wilcoxon test: *p* = 0.011) (Figure 2A), with READ tumors expressing overall higher levels of *MACC1* compared to those of the COAD cohort (median: COAD tumors = 10.49; READ tumors = 10.74, Wilcoxon test *p* = 0.013). Of note, the analysis also revealed higher variability of *MACC1* expression among COAD tumor samples, with notably some outliers with very low values (<8) (COAD ranged from 5.58 to 12.55; READ: 8.99 to 13.07). Similar observations were made in the data set of Wang et al. [27] dataset (Appendix A).

We next confirmed that, as primarily observed in the Oncotrack cohort data, *MACC1* was subjected to SCNAs (SCNA ≠ 0) in 62% of the tumors of the COAD-READ cohort (N = 233/374) and that increased *MACC1* SCNA correlated with increased mRNA expression (median: SCNA (−1) = 9.36, SCNA (0) = 10.17, SCNA (1) = 10.81, SCNA (2) = 12.2, Wilcoxon test: *p* < 2.2 × 10^−16^) (Appendix A).

We tested the association of *MACC1* expression with clinical and pathological parameters. We found that *MACC1* expression levels varied depending on the tumor localization with an expression that was higher in tumors from the distal part than the proximal (median: cecum = 10.18; ascending colon = 10.54, transverse colon = 10.49, descending colon = 10.46, sigmoid colon = 10.8, rectum = 10.7, Anova test: *p* = 5.5 × 10^−5^) (Figure 2B). Tumors with highest *MACC1* expression (>11) were mostly localized in descending colon and rectum, particularly at the sigmoid and recto-sigmoid junction (median: sigmoid-rectosigmoid = 10.8; others = 10.46, Wilcoxon test: *p* = 3.9 × 10^−5^ (Appendix A). By contrast, among tumors localized in the cecum, there was a subset (n = 7/76) with very low *MACC1* expression (<8) that was not seen in tumors localized in the other anatomic sites. In this cohort, high *MACC1* expression was associated with higher tumor stages (median: stage I–II = 10.45, stage III–IV = 10.62, Wilcoxon test: *p* = 0.013) (Figure 2C) and nodal invasion (median: N0 = 10.49; N1 = 10.62, Wilcoxon test: *p* = 0.03) (Figure 2D), association with metastatic status was only modest (median: M0 = 10.54, M1 = 10.77, Wilcoxon test: *p* = 0.26) (Figure 2E). No significant association was found between *MACC1* levels and size of primary tumors (Figure 2F).

#### 3.3.2. Association with Tumor Subtypes and Molecular Characteristics

We next analyzed *MACC1* SCNA and expression in the context of CRC subtypes. For this, we used the sample subtype annotations provided by Liu et al. [28], that were available for 243/638 tumor samples of the COAD-READ cohort (Appendix A). In this tumor subset, we validated that high *MACC1* expression correlated with increased SCNA and occur preferentially in tumors located in the distal part of the colon (Figure 3A). We noticed, additionally, that the gene was rarely mutated (COAD: 24/367—7%; READ: 4/121—3%).

The genomic subtypes of these CRC samples were microsatellite instable (MSI: 36; 14.8%), hypermutated-single nucleotide variants (HM-SNV: 3; 1.2%; hypermutated defined by mutation density >10 per megabase (Mb)), genomic stable (GS: 33; 13.6%) and chromosome unstable (CIN: 171; 70.4%) (Figure 3B). High *MACC1* SCNAs and high expression levels were mostly observed for CIN tumors, whereas MSI, HM-SNVs and GS tumors with normal ploidy had lower *MACC1* expression levels (median: CIN = 10.8, GS = 10.29, HM-SNV = 9.92, MSI = 9.65, Wilcoxon test: CIN vs. MSI: *p* = 1.4 × 10^−10^, CIN vs. GS: *p* = 0.0001, HM-SNV was excluded from the analysis due to small sample size) (Figure 3A, C). When CIN tumors are subclassified as Focal or Broad depending on the length of the gene copy number alterations [28], CIN tumors with *MACC1* SCNA are preferentially classified into the CIN-Focal group (71%) than in the CIN-Broad (29%). However, the difference of *MACC1* mRNA expression between both the CIN-Broad and the CIN-Focal groups was not significant (Appendix A), Wilcoxon test *p* = 0.53). In contrast, the subset of tumors with a very low *MACC1* expression belonged mostly to the MSI tumors having normal ploidy (Figure 3A,C). The low *MACC1* expression was also associated with high CpG island methylator phenotype (CIMP-H), predominantly observed in MSI tumors [32], compared to CIMP-low (CIMP-L) or non-CIMP (median: CIMP-H = 10.18, CRC CIMP-L = 10.54, non-CIMP = 10.77; Wilcoxon test: *p* = 0.0035 and *p* = 0.0046, respectively) (Appendix A).

To obtain more insight into the genetic background and aggressive phenotype associated with *MACC1* expression, we tested whether certain mutated genes can be associated with its expression levels (cohort of 243 tumors with both transcriptomic and mutation data). Focusing first on CRC-related cancer genes, we found only some weak associations. *MACC1* expression was lower in tumors mutated for *KRAS* (median: mutated = 10.52; wild type = 10.76; Wilcoxon *p* = 0.005), irrespective of G12 or G13 mutation types and *POLE* (catalytic subunit of DNA polymerase epsilon; median: mutated = 10.02; wild type = 10.61; Wilcoxon test: *p* = 0.042) genes, whereas no other significant association was found for the other genes (Figure 3A, Appendix A).

*MACC1* was next analyzed in the context of the transcriptomic-based consensus molecular subtypes (CMSs) which were established based on the tumor gene expression [33]. In the cohort reported [28], CRCs classified into CMS1 35/243 (microsatellite instability immune, 14.4%), CMS2 80/243 (canonical, 32.9%), CMS3 30/243 (metabolic, 12.3%) and CMS4 63/243 (mesenchymal, 25.9%), and CMSx 35/243 (indeterminate or undefined: 14.4%) (Figure 3D). *MACC1* expression levels were associated with the tumor CMS subtypes (median: CMS1 = 10.1, CMS2 = 10.8, CMS3 = 10.2, CMS4 = 10.6, Kruskal–Wallis test *p* = 1.6 × 10^−6^) (Figure 3A,E). *MACC1* expression was higher in tumors of the CMS2 subtype compared to CMS1 and CMS3 (Wilcoxon test *p* = 1.5 × 10^−6^ and *p* = 0.003, respectively) but not CMS4. Similarly, *MACC1* was higher in CMS4 subtype than in CMS1 and CMS3 (Wilcoxon test *p* = 5.3 × 10^−5^ and *p* = 0.025, respectively). *MACC1* levels tended to be lower in CMS1 than in CMS3 subtype (Wilcoxon test *p* = 0.1). In summary, we showed here that tumors with high *MACC1* expression and increased SCNAs were mostly seen in CIN tumors of the CMS2 subtype, which is the epithelial type with marked WNT activation, and of the CMS4 subtype defined as mesenchymal with prominent transforming growth factor activation, stroma invasion and angiogenesis. Tumors of the CMS3 subtype were frequently GS with overall lower *MACC1* expression levels. Finally, the CMS1 subtype aggregated mostly MSI, HM-SNV and CIMP-H tumors with normal ploidy and frequently very low *MACC1* mRNA expression.

We also investigated *MACC1* expression with the immune contexture of the tumors. We used the quanTIseq deconvolution method [34] to quantify the fractions of immune cell types from bulk COAD-READ RNA-Seq tumor samples (N = 380). Each sample was analyzed for respective content of ten immune cell types including B cells, M1 and M2 macrophages, monocytes, neutrophils, natural killer (NK), CD4+ and CD8+ T cells, regulatory T cells (Tregs) and dendritic cells, versus the content of any other cell type present in the sample. The overall immune cell content ranged from 12% to 60% (median = 26%). Among the immune cells, neutrophils have the highest percentage in COAD-READ tumors (median = 9.1%), followed by M1 macrophages (median = 4.7%), while monocytes were not detected. High *MACC1* mRNA levels positively correlated with non-immune cell content (rho = 0.19, *p* = 0.0002) and negatively with sample overall immune cell content (rho = −0.18, *p* = 0.0003) (Appendix A). *MACC1* levels mostly inversely correlated with the proportion of CD8+ T cells (rho = −0.21, *p* = 2.43 × 10^−5^), M1 macrophages (rho = −0.18, *p* = 0.0004) and Treg cells (rho = −0.14, *p* = 0.005)

#### 3.3.3. Association with CRC Patient DFS and OS

In the COAD-READ cohort, we tested the association between *MACC1* SCNA, elevated mRNA expression and CRC patients DFS and OS. For that, we merged clinical and molecular data that were available for 372 patients. The clinical data were obtained from firehose (http://firebrowse.org/, accessed on 9 May 2018). Curated DFS and OS information were collected from the National Cancer Institute website at: https://gdc-portal.nci.nih.gov/legacy-archive/, accessed on 12 July 2021 [35]. *MACC1* cut-offs were set at: SCNA > 0 and mRNA (log_2_) > 10.57 (corresponding to the median of *MACC1* expression) to dichotomized CRC tumors.

In univariate analysis, we found that *MACC1* expression but not SCNA was significantly associated with patient DFS (*MACC1* > 10.57: HR: 1.88; 95% CI [1.27 to 2.88]; Wald test *p* = 0.003) and OS (*MACC1* > 10.57: HR: 1.77; 95%CI [1.14 to 2.74]; Wald test *p* = 0.01) (Figure 4A). Large tumor sizes, advanced cancer stages, lymph node invasion and metastatic status were, as expected, strong predictors of shorter DFS and OS in this patient cohort (all: *p* < 0.05). Patient age was also associated with shorter OS (*p* < 0.05) but not DFS.

The association of *MACC1* expression with patient DFS and OS was further assessed by multivariate analysis against tumor size and stage, nodal and metastatic status. With this parameter combination, *MACC1* expression did not retain significance for association with both DFS and OS (*p* = 0.1 and *p* = 0.18, respectively); the patient metastatic status remained the only parameter significantly associated with DFS and OS (*p* = 0.002 and *p* = 0.03, respectively) (Figure 4A). By removing metastatic status from the analysis, we showed that high *MACC1* expression and high tumor size were independent predictors of reduced DFS (*p* = 0.02 and *p* = 0.03, respectively) (Appendix A). For OS, while *MACC1* expression showed a similar trend than for DFS (*p* = 0.11), only the tumor stage was statistically significant (*p* = 0.001).

In Kaplan–Meier and log-rank testing analyses combining both *MACC1* SCNA and mRNA expression, we delineated four subsets of CRC patients with distinct DFS (*p* = 0.01, Figure 4B) and for OS (*p* = 0.05, Figure 4C). Patients with tumors having both *MACC1* SCNAs and high mRNA expression presented the shortest outcome with a median DFS and OS of 40 and 56 months, respectively. Those having high *MACC1* mRNA expression without SCNA had also a short outcome with median DFS and OS of 59 and 62 months, respectively. Interestingly, we noticed that tumors with *MACC1* SCNA and high expression predict faster disease progression and death when compared to tumors having high mRNA expression but no SCNA (Figure 4B,C). In contrast, the patients carrying tumors without *MACC1* SCNA and with low mRNA levels had longer DFS and OS of 75 and 100 months, respectively. There was also a subset of tumors with *MACC1* SCNA that was not accompanied by a particularly high mRNA expression. These patients also had a longer DFS of 109 months; for OS, the curve did not drop to a probability below 50%, thus the median cannot be computed.

## 4. Discussion

Here, we report a novel fundamental *MACC1* regulatory layer in the context of *MACC1*-induced clinical consequences in CRC. We demonstrate (i) that *MACC1* gene expression is strongly dependent on gene SCNA, determining not only *MACC1* gene expression levels, but also *MACC1* dependent subsequent metastasis formation. Further, we show (ii) that elevated *MACC1* expression, driven by CIN and gene SCNA, is associated with molecular subtype, especially with CMS2 and CMS4.

The gene *MACC1* is a key regulator of the HGF/c-Met pathway and its overexpression is causative for cancer cell proliferation, colony formation, dissemination, migration, and invasiveness in cell culture and for tumorigenesis, tumor progression and metastasis formation in several mouse models [4,5,19,36]. This gene was discovered first in CRC but is meanwhile acknowledged as a prognostic and predictive biomarker for more than 20 solid cancer entities [4,6]. *MACC1* expression level is the main parameter, determining all these phenotypes, which can be determined at RNA and/or protein level using different technologies in tumor tissue and/or patient blood (cell-free RNA, circulating tumor cells). However, the actual cause of high *MACC1* expression in patient tumors or blood has still to be elucidated.

We started this study with correlation analyses between *MACC1* gene SCNA and mRNA expression levels in independent patient datasets, the Oncotrack cohort and the Charité cohort. We found strong positive associations of high *MACC1* gene copy number and increased *MACC1* expression levels.

We further validated that *MACC1* SCNA and high mRNA levels were accompanied by high protein levels by using the CPTAC-2 prospective cohort [30]. Overall, most tumors of this cohort having a *MACC1* SCNA gain showed also high protein levels. Only a few samples showed *MACC1* gene gain, high mRNA expression but low protein levels. It could be that for these few tumors, the regulation occurred at another level in addition to the described ones: A large body of reports describe post-transcriptional regulations of *MACC1* expression by miRNA, lnc-RNA and circRNA [5,16]. The identification of the *MACC1* gene promoter enabled the identification of transcription factors and respective transcription factor binding sites regulating the transcription of the metastasis inducer *MACC1* [14,15].

More importantly, we tested for the correlation of *MACC1* SCNA with patient metastasis. Additionally, here we found strong positive associations of high *MACC1* gene copy number with increased metastasis frequency and worse patient outcomes. Thus, determining *MACC1* SCNA not only provides a basis for predicting mRNA expression levels, but also represents a novel criterion to predict CRC aggressiveness, potentially determining the molecular risk for metastasis, and for selecting intervention strategies. In this regard, we focus on gene-specific approaches (e.g., [4,33] or pharmacological options inhibiting transcriptional *MACC1* overexpression, e.g., statins, such as lovastatin or potentially rottlerin [34]. Taken together, assessment of *MACC1* expression by SCNA, or in combination with the mRNA level, might contribute to augmenting patient stratification strategies, but may also help to justify and broaden the use of, e.g., anti-metastatic treatment options.

We showed next by using the TCGA COAD-READ cohort, that increased *MACC1* SCNA and expression occurred in tumors with chromosomal instability (CIN molecular subtype) for which CRC patients have, in general, a less favorable outcome compared to MSI or GS tumor subtypes [37]. The context of copy number gain and overexpression of *MACC1* was also analyzed by Galimi and colleagues [38] in a cohort of 103 consecutive metastasized colorectal carcinomas. They report a recurrent specific gain of the p-arm of chromosome 7 (where *MACC1* is located) and a positive correlation of *MACC1* copy number gain with unfavorable pathologic features. *MACC1* showed preferential expression in highly aggressive and high-grade tumors, being more expressed in multiple vs. single metastases, larger versus smaller tumors, and in the presence of intravascular metastatic cells and metastatic emboli. They conclude that copy number gain and overexpression of *MACC1* correlated with pathologic attributes of tumor evolution in the context of liver metastases from CRC. Shimokava and colleagues demonstrated this role of *MACC1* in 146 consecutive patients who underwent a complete resection for stage I lung adenocarcinoma [39]. They report a median *MACC1* copy number of 3.0 in patients with tumor recurrence and 1.4 in patients without recurrence. This finding supports the link of *MACC1* copy number gain and tumor evolution also for lung cancer.

With the analysis of the TCGA COAD-READ cohort, we confirmed the elevated expression of *MACC1* mRNA in tumors as compared to normal tissues. *MACC1* high expression occurred preferentially in colon tumors at the distal sites. As we and others reported previously [40,41], there is an association between high *MACC1* mRNA levels and clinical markers of aggressiveness, such as higher tumor stages or nodal invasion. Some variability was, however, seen regarding the strength of the statistical associations. These differences could be due to different methods for MACC1 assessment and the heterogeneity of *MACC1* expression within the tumors that have an impact on its quantification [40,41]. In our study, *MACC1* mRNA levels were also shown to predict the patients’ DFS and OS in a univariate analysis. In line with our previous observations [40,41], *MACC1* levels showed dependency with the patient’s metastatic status in a multivariate analysis. Its independent prognostic value was retained only by removing the metastatic status from the variables.

In the analyses it was not possible to validate *MACC1* SCNA as a prognostic marker, using the data from the TCGA COAD-READ cohort. In the study with the Oncotrack cohort (Figure 1A), *MACC1* mRNA levels were directly dependent on the degree of *MACC1* SCNA, this most probably related to the degree of polyploidy of the tumors. We found samples displaying weak SCNA (3N) along with lower mRNA expression whereas some others (4N or more) with high mRNA expression. Unfortunately, the SNP 6.0 data publicly available for the TCGA cohorts were analyzed with the GISTIC method which fails to discriminate between samples exhibiting low to high gene SCNA (samples with low to high SCNA all categorized SCNA = 1). To use *MACC1* SCNA as a biomarker, methods, such as PICNIC [42] for the SNP 6.0 microarray analysis or a ddPCR assay should be used to better discriminate between low and high SCNA.

Further, CRC is classified into four CMSs with distinct molecular and biological features: CMS1 (microsatellite instability immune; enriched for MSI tumors and *BRAF*-mutations), CMS2 (canonical; epithelial characteristics with marked WNT and MYC signaling and high CIN), CMS3 (metabolic; epithelial features but less CIN, enriched for *KRAS* mutations, evident metabolic dysregulation), and CMS4 (mesenchymal; prominent TGF-β activation, stromal invasion, angiogenesis, inflammatory, immunosuppressive phenotype) [33]. When applied to CRC translational research, these subtypes will impact disease prognosis, therapeutic treatment options and improve clinical outcomes [43]. Here we used TCGA data and in silico approaches for a deep and integrative analysis of *MACC1* expression modalities in CRCs. We found that elevated *MACC1* expression, driven by CIN and SCNA, is associated with molecular subtype, especially with CMS2 and CMS4.

Deregulation of the Wnt/β-catenin pathway resulting in neoplastic transformation and tumor progression is associated with overexpression of target genes, such as c-Myc or cyclin D1. The interplay of MACC1 and Wnt/β-catenin signaling, meanwhile repeatedly studied, was also supported by Meng et al., where the functional relationship of β-catenin, *MET* expression, and *MACC1* was shown to be decisive for tumor growth and metastasis formation in nasopharyngeal carcinoma [44]. MACC1 also promoted carcinogenesis of CRC via β-catenin signaling pathway [45], suppression of miR-338-3p up-regulated *MACC1*, β-catenin and *VEGF* expression leading to angiogenesis in hepatocellular carcinoma [46]. *MACC1* knockdown effectively inhibited proliferation and promoted apoptosis of lung adenocarcinoma cells by regulating the β-catenin pathway (including c-myc) [47], and DBC1 (BMP/retinoic acid inducible neural specific (1), which play a key role in CRC progression through Wnt/β-catenin-MACC1 signaling axis [48]. Results from our group showed in vil-MACC1/ApcMin transgenic mice the interplay of Wnt signaling and MACC1 action for CRC tumor formation [5]. Transcriptomic analysis demonstrated increased Wnt and pluripotency signaling and the enhancement of the pluripotency markers Oct4 and Nanog in the tumors of vil-MACC1/ApcMin mice (vs. ApcMin mice). In addition, the MACC1 and c-Myc relationship was further substantiated showing a *c-Myc* increase by *MACC1* overexpression in CRC [45], a correlation of *MACC1*/*c-Myc* expression in endometrial carcinoma [49], and binding of miR-384 and miR-145-3p miRNAs to MACC1-AS1 thereby altering cell growth phenotype through increased expression of *c-Myc* mRNAs and *PTN* (pleitrophin, secreted heparin-binding growth factor [50]). Taken together, for tumors of CMS2, combinatorial approaches simultaneously targeting Wnt signaling (β-catenin, c-Myc) and *MACC1* might be beneficial for CRC patients.

The causative connection of *MACC1* and CMS4 has been demonstrated by a multitude of reports, addressing TGF-β activation, stromal invasion, angiogenesis and an inflammatory phenotype. For instance, TGF-β1 secretion by mesenchymal stem cells activated SMAD2/3 through TGF-β receptors and induced lncRNA *MACC1-AS1* expression in gastric cancer cells, which promoted stemness and chemoresistance [51], and *MACC1-AS1* positively regulated TGF-ß1 expression, resulting in increased invasion and migration rates in hepatocellular cancer cells [52].

*MACC1*-induced stromal invasion has been shown in a variety of studies for CRC [4,6], and in CRC animal models [5]. Elevated *ORAI1* and *STIM1* expression upregulated *MACC1* expression and promoted tumor cell proliferation, metabolism, migration, and invasion in human gastric cancer.

Further, the causal involvement of MACC1 in angiogenesis was reported in several studies, e.g., for CRC, cervical, gastric, and hepatocellular cancer, for cholangiocarcinoma, brain microvascular endothelial cells, osteosarcoma, and oral squamous cell carcinoma [14]. MACC1 and VEGF family members contribute to vasculogenesis, angiogenesis and vasculogenic mimicry, and are upregulated in many cancer types, correlating to tumor stage and progression. MACC1 induced VEGF-C/VEGF-D in gastric cancer, and VEGF-A in gastric cancer and cholangiocarcinomas [14,53].

Linking MACC1 and inflammation was demonstrated by Harpaz et al. showing a *MACC1* expression increase from inflammatory bowel disease (IBD)-associated colitis to dysplasia to adenocarcinoma suggesting that *MACC1* is strongly associated with conventional tumorigenesis of colitis-associated CRC [54]. For CRC, we reported that pro-inflammatory TNF-α and IFN-γ promote tumor growth and metastasis via induction of *MACC1* [55]. These examples underline that elevated *MACC1* expression is not only substantially driven by chromosomal instability and SCNA but is also importantly linked to molecular subtype.

In our study, we also showed that high *MACC1* mRNA levels inversely correlated with the overall content of immune cells in the CRC samples. In contrast with MSI+ tumors, those with chromosomal instability, such as the ones with *MACC1* SCNA are known to have lower levels of mutations, fewer neoantigens on the cell surface and overall less inflammatory components and lower response to immunotherapy [56]. Supporting this, we also observed an inverse correlation between *MACC1* levels and levels of CD8+ T cells that is a marker for sensitivity to immunotherapy [57]. It could suggest that complementary to already existing methods, *MACC1* SCNA or its expression levels could be used as a decisional marker for immunotherapy treatment in CRC.

## 5. Conclusions

Overall, here we report novel insights into the contexture of MACC1 expression in CRC. We demonstrate that high expression levels of MACC1 in CRC tumors, associated with metastasis and worse patient outcome, is frequently caused by increased gene copy number. High MACC1 expression, CIN, DNA copy number gains, and CMSs, potentially define the molecular risk for cancer metastasis and might serve for refined diagnostic and patient tailored treatment decisions.

## Figures and Tables

**Figure 1 cancers-14-01749-f001:**
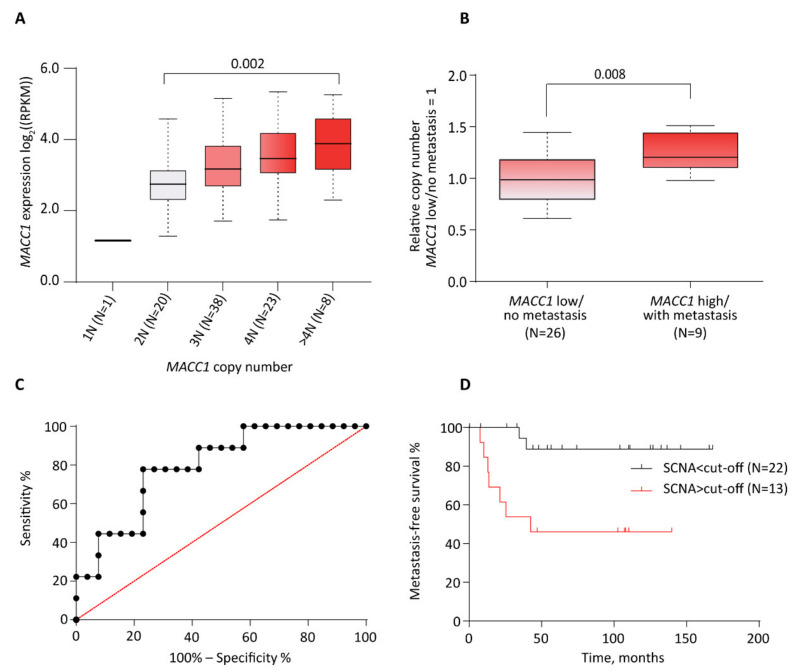
Overlay of copy number, expression of MACC1, and metastasis in the CRC OncoTrack cohort and Charité metastasis cohort. (**A**), Box plot shows expression distribution of samples with different copy number estimates for MACC1. The copy numbers are indicated below each box (xN). The number of samples per group is shown in brackets. (**B**), MACC1 SCNA in patients with and without metastasis. Patients with known metastasis status and MACC1 gene expression level were assessed for MACC1 copy number. Patients with low MACC1 gene expression and no metastasis showed less genomic MACC1 copies compared to patients with unfavorable high MACC1 gene expression and diagnosed metastasis. (**C**), Receiver operating characteristic (ROC) curve to define the cut-off value for MACC1 gene copy number discriminating patients in groups above and below this threshold. The maximum of the Youden-index (J = Sensitivity + specificity − 100) was used to identify this value. (**D**), Kaplan-Meier analysis showing CRC patient metastasis-free survival stratified by tumor MACC1 gene copy number (SCNA > cut-off vs. SCNA > cut-off). The median was used as the cut-off point for *MACC1* expression.

**Figure 2 cancers-14-01749-f002:**
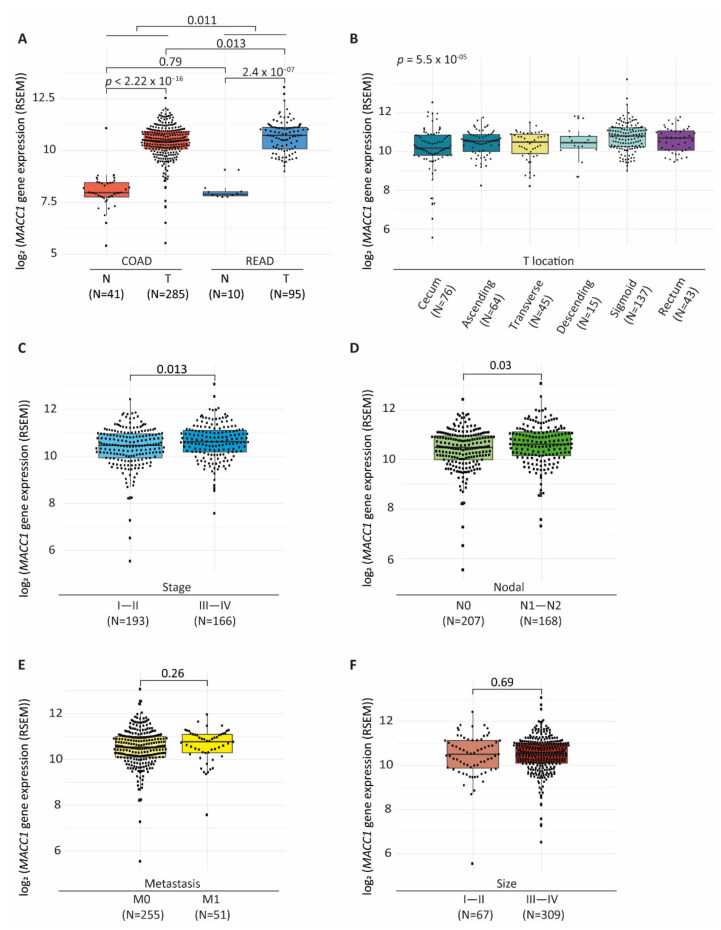
MACC1 expression in the COAD-READ dataset. (**A**), Box plot showing MACC1 mRNA expression values of the COAD and READ samples stratified by normal or tumor tissue types. (**B**–**F**), MACC1 mRNA expression by tumor (**B**), localization, (**C**), stage, (**D**), patient nodal invasion, (**E**), patient metastatic status, (**F**), size. Box plots (**A**–**F**), MACC1 data are RNA-Seq data RSEM-log2 transformed, horizontal bars indicate median values, boxes are interquartile range, whiskers indicate values within 1.5 times interquartile range. The number of samples (N) per group is given in brackets.

**Figure 3 cancers-14-01749-f003:**
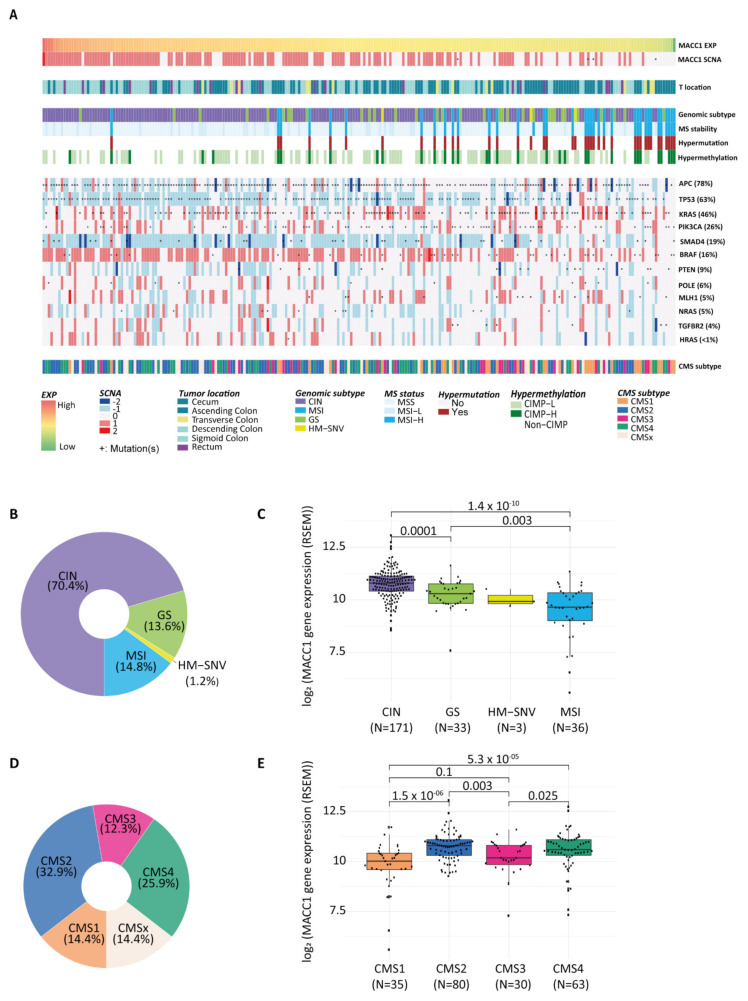
MACC1 mRNA expression and CRC molecular subtypes. (**A**), Landscape showing the association of MACC1 SCNA and expression according major colorectal-cancer-associated subtypes including MSI status, genomic subtypes, hypermutation status, hypermethylation, consensus molecular subtypes (CMS), tumor location and genes commonly mutated in CRC. (**B**), Donut chart representing the percentage distribution of each genomic molecular subtype in the Liu et al. COAD-READ tumor subset. (**C**), Box plot of MACC1 mRNA expression by tumor molecular subtypes (genomic). HM-SNV was excluded from the analysis due to small sample size. (**D**), Donut chart showing the percentage of tumors classified according to the consensus molecular subtypes (CMS 1–4), CMSx are mixed or indeterminate. (**E**), Box plot showing the distribution of the MACC1 mRNA expression in the different CMS tumor subtypes. (**C**,**E**), horizontal bars indicate median values, boxes are interquartile range, whiskers indicate values within 1.5 times interquartile range.

**Figure 4 cancers-14-01749-f004:**
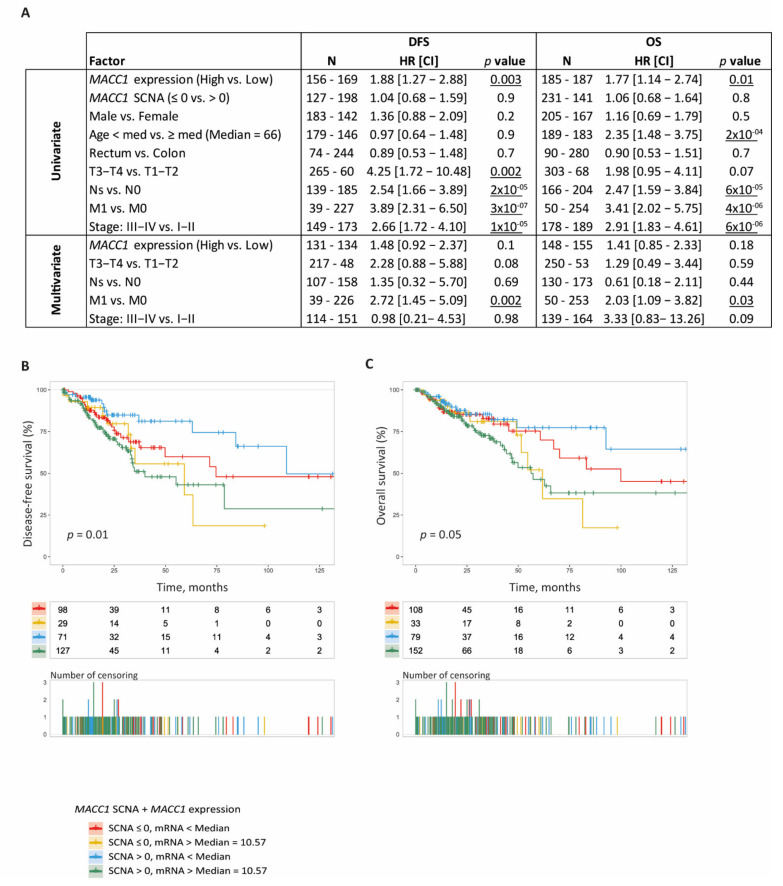
MACC1 SCNA and expression and CRC patient DFS and OS. (**A**), Univariate and multivariate Cox analysis (DFS and OS) in COAD-READ TCGA patients (N = 218). HR: Hazard ratio; CI: Confidence interval. N represents the number of cases in each group. Wald test was used to identify the risk factors with patient DFS and OS. Significance setup at *p* < 0.05. (**B**,**C**), Combined Kaplan-Meier analysis of MACC1 status (SCNA and mRNA). (**B**), DFS according to the MACC1 status, (**C**), OS according to the MACC1 status.

## Data Availability

Data are available on reasonable request via the corresponding author. The datasets related to the analyses performed in this study including links to the publicly archived datasets can be found in Appendix A.

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
