# Peer review of "Elevated MACC1 Expression in Colorectal Cancer Is Driven by Chromosomal Instability and Is Associated with Molecular Subtype and Worse Patient Survival"

_cancers, 2022, doi:10.3390/cancers14071749_

Round 1

Reviewer 1 Report

Vuaroqueaux et al. performed a study investigating colorectal cancer using SCN- and RNA Expression analysis to further evaluate MACC1 in the context of other molecular aberrations. The enrolled 141 cases of two cohorts and additionally analyzed TCGA data

Basically, the study was carried out in an elaborate manner, the methods used appear adequate and the illustrations were carefully prepared.

Nevertheless, it was difficult to follow the study. The results are in some ways contradictory, which is not sufficiently acknowledged in the discussion. Furthermore, I am not convinced of the relevance of this biomarker, which was initially well published. Since then, about 300 articles have been written, which is not a huge number, more still about 40 articles originate from the group of authors of this manuscript. Therefore, there is probably only a moderate interest in the scientific community.

  • The hypothesis stated in the header could not be sufficiently confirmed by the data shown, so the header should be changed.
  • Point 3.1: Only mRNA-Expression Levels were used to correlate SCNAs. Did the authors also check protein expression, as protein expression itself is highly regulated and could be different from mRNA-expression levels.
  • Figure 1B and D: N-Number of the two groups should be added to the graph.
  • Figure 1D: For an easier and better understanding, the definition of the used cut-off should be added to the figure legends.
  • Figure 2: The differences between the tumor subtypes are really small, even if they are statistically significant. Is there a real clinical consequence expected? This should be discussed in more detail.
  • Figure 3C: N-Number of HM-SNV (n=3) is very low, compared to the other groups. Therefore, these data are not very robust and should be excluded from the statistics.
  • Figure 4A: In the multivariate analysis, only MACC1 RNA expression seems to have an impact on patient survival, which means that MACC1 RNA expression alone is important, regardless of SCNA? Please discuss this in more detail.
  • Figure 4C/ Point 3.6:
    • One patient group (blue) shows a SCNA but no higher mRNA expression levels. In Figure 1 the authors postulate a clear correlation between SCNA and mRNA expression. So, these data contradicting each other. How it can be explained, that there are patients with a higher copy number of the gene, but with no higher mRNA expression?
    • The blue patient group seems to have the best clinical outcome, even better than patients with no SCNA and no higher mRNA-Expression. It seems that the SCNA is not the critical factor, as the authors postulate. The actual MACC1 expression seems to be more relevant.
  • Table S2 is very confusing and should be revised. According to Table S2 Figure 1D is only based on 17 patients?
  • Overall, the work seems very confusing due to the use of different collectives. It should be better structured, e.g. with an overview chart of the different collectives and the different methods used. This would significantly improve the comprehensibility of the paper.
  • Typos in the heading of the S3 table.
  • Discussion:
    • The authors should discuss their own data more in detail.
    • The clinical relevance of the data should be discussed more in detail.
    • Limitations of the study should be discussed.

Reviewer 2 Report

In this article, authors performed mostly in silico of public or already published transcriptomic/genomic data from colorectal cancer samples, to evaluate MACC1 SCNA or mRNA expression levels and their association with metastasis, stage, tumor subtype, molecular characteristics, immune infiltration, and patient survival. Even when the association of MACC1 and colorectal cancer has been already studied by others, here authors provide further information using large cohort of patients from different sources. The article is of good quality, and results can have significant impact on the development of patient-tailored treatment decision. Materials and methods are precisely described. Here my comments:

Comment #1: Authors should indicate the purpose of the study in the abstract.

Comment #2: All non-standard abbreviations/acronyms should be written out in full on first use, also in the abstract (e.g. SCNA, COAD, READ).

Comment #3. In line 333, MACC1 should be in italics.

Comment #4. It is a bit confused when authors stated that high MACC1 mRNA levels occurred preferentially in “cold tumors” (lines 353-354), while were inversely correlated with Treg cells, a key cell type of “cold tumors”. Please, clarify the differences between “hot tumors” and “cold tumors”, regarding immune components, and deeply discuss this data.

Comment #5: In Figure 1A, authors demonstrated that MACC1 expression levels were copy number dependent. In Figure 1B, they showed significant higher somatic copy number alteration (SCNA) in the MACC1 high/with metastasis group vs MACC1 low/no metastasis group. In order to properly define the correlation between MACC1 expression, MACC1 SCNA and metastasis, authors should analyze the correlation among the 3 variables, or compare MACC1 SCNA between 4 groups: a) MACC1 high/with metastasis group, b) MACC1 high/no metastasis, c) MACC1 low/no metastasis group, d) MACC1 low/with metastasis group; similar analysis was performed for Figure 4B and 4C, when compared overall survival and disease free survival with MACC1 expression and SCNA.

Comment #6. In Figure 2, along with MACC1 expression, it would be interesting to see the correlation between MACC1 SCNA and the clinical/pathological parameters evaluated here: normal vs tumor expression, tumor localization, stage, patient nodal invasion, patient metastasis status, tumor size.
